# HetGSMOTE: Oversampling for Heterogeneous Graphs

Adhilsha Ansad[1], Deependra Singh[1], Rucha Bhalchandra Joshi[*2], and Subhankar Mishra[†1]

[1]National Institute of Science Education and Research, Bhubaneswar, India
[2]The Cyprus Institute, Nicosia, Cyprus
{adhilsha.a, deependra.singh}@niser.ac.in, r.joshi@cmi.ac.cy, smishra@niser.ac.in

## Abstract

Graph Neural Networks (GNNs) have proven effective for learning from graph structured data, with heterogeneous graphs (HetGs) gaining particular prominence for their ability to model diverse real world systems through multiple node and edge types. However, class imbalance where certain node classes are significantly underrepresented presents a critical challenge for node classification tasks on HetGs, as traditional learning approaches fail to adequately handle minority classes. This work introduces HetGSMOTE, a novel oversampling framework that extends SMOTE-based techniques to heterogeneous graph settings by systematically incorporating node-type, edge-type, and metapath information into the synthetic sample generation process. HetGSMOTE operates by constructing a content-aggregated and neighbor-type-aggregated embedding space through a base model, then generating synthetic minority nodes while training specialized edge generators for each node type to preserve essential relational structures. Through comprehensive experiments across multiple benchmark datasets and base models, we demonstrate that HetGSMOTE consistently outperforms existing baseline methods, achieving substantial improvements in classification performance under various imbalance scenarios, particularly in extreme imbalance cases while maintaining broad compatibility across different heterogeneous graph neural network architectures. We release our code and data preparations at github.com/smlab-niser/hetgsmote.

## 1 Introduction

Graph-based learning and Graph Neural Networks (GNNs) have garnered significant attention for their capacity to model intricate relationships and dependencies within structured data [1]. Among these graph-based structures, heterogeneous graphs (HetGs) represent a particularly important class that involves multiple types of nodes and edges. HetGs are well-suited for modeling real-world data where di-

verse entities and relationships must be represented distinctly. Examples include recommender systems, social networks, and bibliographic networks [2, 3], where entities such as authors, papers, and venues interact through different relationship types. These diverse entities and relationships encode rich, complex information, with nodes often containing both structured and unstructured content, making downstream tasks like node classification [4, 5] and link prediction [6, 7] particularly important.

While Heterogeneous Graph Neural Networks (HGNNs) [5] excel in node classification tasks, they typically assume balanced class distributions to achieve consistent performance across all classes. However, many real-world applications exhibit significant class imbalance, where certain classes have substantially fewer instances than others [6, 8]. This imbalance leads to suboptimal performance when using traditional baseline models, necessitating specialized imbalanced learning techniques such as oversampling for heterogeneous graphs.

Synthetic Minority Oversampling Technique (SMOTE) has proven effective for balancing class distributions across various domains, including homogeneous graphs [9]. Different SMOTE variants have been developed to address imbalance through various strategies. Some focus on specific spaces within minority classes [10, 11], while others target difficult-to-learn regions [12–14]. As these oversampling techniques continue to evolve, their adaptation to heterogeneous graphs requires careful consideration of the unique structural properties involved. However, its direct application to heterogeneous graph data presents unique challenges: type-specific semantics, edge heterogeneity, rich contextual information from diverse metapaths, and non-shared feature spaces.

To address these challenges, we present the *HetGSMOTE* framework as shown in Figure 1, which extends the GraphSMOTE approach [9] from homogeneous to heterogeneous graphs. Our framework constructs a content-aggregated and neighbor-type-aggregated embedding space that encodes node similarities, facilitating the generation of synthetic samples that preserve contextual relationships within the graph. Additionally, HetGSMOTE trains specialized edge generators for different node types to

---

*This work was performed while the author was affiliated with NISER[1].

†Corresponding Author.

Proceedings of the 7th Northern Lights Deep Learning Conference (NLDL), PMLR 307, 2026.

model relational information between nodes, ensuring that synthesized samples retain essential structural characteristics. This approach generates synthetic sample representations in safe regions of the embedding space, where source samples are less prone to producing noisy synthetic data.

The main contributions of this paper are:

1. We address the class imbalance problem in heterogeneous graphs and propose a comprehensive solution with broad real-world applicability.

2. We extend the GraphSMOTE oversampling approach into a complete framework for heterogeneous graphs through our novel HetGSMOTE strategy.

3. We demonstrate our approach's efficacy through comprehensive experiments across diverse settings, datasets, and base models, showcasing superior performance compared to existing baseline methods.

## 2  Related Works

Class imbalance, where one class significantly outnumbers another, leads to biased models and poor generalization, evident in tasks such as fraud detection, rare disease identification, and bot recognition. Addressing this issue involves algorithm-level, data-level, and hybrid strategies [9]. Data-level methods, such as oversampling and data augmentation, increase minority class samples. Algorithm-level approaches include cost-sensitive techniques, ensemble learning, and threshold adjustments [15, 16]. Hybrid methods combine these strategies, e.g., classifier-specific models [8].

Oversampling generates synthetic minority samples with methods like SMOTE, which interpolates between samples. Approaches like DBSMOTE[10], and k-means SMOTE[11], focus on generating synthetic samples within the minority class space with smaller scales. Methods like borderline-SMOTE[12], ADASYN[13], and Adaptive-SMOTE[14] also generates samples in difficult regions within minority class. These improvements help mitigate overgeneralization by filtering out potential noise or by strategically generating additional samples within specific regions of the minority class. A newer technique, NaNG-SMOTE[17], addresses the same obstacles by using a natural neighborhood graph and subgraph cores of the minority class to generate synthetic samples while filtering noise based on edge characteristics. Oversampling has proven effective in numerous machine learning domains, addressing the issue of limited minority data [8] and improving model performance.

Oversampling techniques for graph data, like GraphSMOTE [9], address class imbalance in homogeneous graphs by generating synthetic nodes and connections. Other methods, such as GraphMixup [18] and Graph-DAO[19], use latent space sampling and semantic relations. Despite progress, SMOTE has found limited applications in graph-based learning. For HetGs, there are even fewer techniques, such as FincGAN [20], an adversarial GAN-based approach, and BARE[21], which leverages student-teacher networks to distill knowledge from real nodes for improved learning. This motivates our work on the extension of GraphSMOTE to heterogeneous graphs.

## 3  Problem Statement

Consider a heterogeneous graph (HetG) denoted as $\mathcal{G} = (V, \mathcal{A}, \mathcal{F})$, where:

- $V = \{v_1, \dots, v_n\}$ is the node set, with $V_t \subseteq V$ representing the subset of nodes of type $t \in \mathcal{T}$, where $\mathcal{T}$ is the set of node types.

- $\mathcal{A} = \{A_{tu} : t, u \in \mathcal{T}\}$ is the set of adjacency matrices, where $A_{tu} \in \{0, 1\}^{n_t \times n_u}$ represents the adjacency matrix between nodes of types $t$ and $u$, with $n_t = |V_t|$ and $n_u = |V_u|$.

- $\mathcal{F} = \{F^t : t \in \mathcal{T}\}$ is the set of node feature matrices, where $F^t \in \mathbb{R}^{n_t \times d}$ contains the $d$-dimensional feature vectors for all nodes of type $t$. Specifically, $F^t[v_j, :] \in \mathbb{R}^{1 \times d}$ represents the feature vector for node $v_j$ of type $t$.

For node classification tasks, we focus on a specific target node type $t^* \in \mathcal{T}$ with associated class labels $Y \in \mathbb{R}^{n_{t^*}}$. In practical scenarios, only a subset of labels is available during training, denoted as $Y' \subseteq Y$ corresponding to the labeled node subset $V' \subseteq V_{t^*}$ where $|V'| \ll |V_{t^*}|$.

We adopt a semi-supervised transductive learning setting where the entire graph structure is available during both training and testing, but only a small fraction of nodes are labeled. These labeled nodes are partitioned into training, validation, and testing sets for the learning process. We adopt the transductive setting to isolate the effect of oversampling on performance without the variability introduced by unseen nodes. Moreover, most heterogeneous datasets are naturally split for transductive evaluation.

**Problem Definition:** Given a heterogeneous graph $\mathcal{G}$ with imbalanced class distribution among labeled nodes $V'$ and their corresponding labels $Y'$, our objective is to:

1. Use the HetGSMOTE framework to generate synthetic minority nodes $V^s$ with corresponding synthetic edges $\mathcal{A}^s$ that preserve the structural and semantic properties of the original graph.

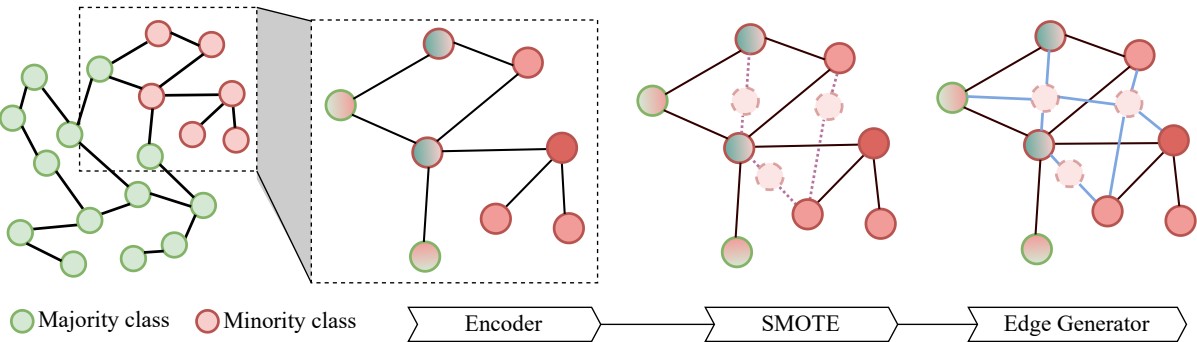

**Figure 1.** Overview of the HetGSMOTE framework pipeline showing the operation of main components.

2. Train a node classifier $f : \mathcal{G} \cup \mathcal{G}^s \to Y$ that achieves balanced performance across both majority and minority classes, where $\mathcal{G}^s = (V^s, \mathcal{A}^s, \mathcal{F}^s)$ represents the synthetic graph components.

The key challenge lies in generating synthetic nodes and edges that maintain the rich semantic relationships encoded in the heterogeneous graph structure while effectively addressing the class imbalance problem for improved classification performance on minority classes.

## 4 Background

This section covers the required background concepts. Details are provided in Appendix A.

**Class Imbalance.** For dataset with classes $\{c_1, \ldots, c_k\}$, imbalance ratio for class $i$ is:

$$\mathrm{IR}_i = \frac{|c_i|}{\max_j |c_j|}$$

**SMOTE** [22]. Generates synthetic minority samples via interpolation:

$$\mathrm{SMOTE}(X_1, X_2) = (1-r)X_1 + rX_2, \quad r \sim \mathrm{Uniform}(0,1)$$

**HGNNs.** Update node $v_j$ of type $t$ as:

$$G^t[v_j, :] = \mathrm{BaseModel}\left(F^t[v_j, :], \mathcal{N}(v_j)\right)$$

Specific models:

- **HetGNN** [5]:
  $G^t[v_j, :] = \frac{1}{|\mathcal{N}^t(v_j)|} \sum_{v \in \mathcal{N}^t(v_j)} \sigma\left(W \cdot F^t[v, :]\right)$

- **HAN** [4]:
  $G^t[v_j, :] = \sum_k \sum_{v \in \mathcal{N}^k(v_j)} \alpha_{v,v_j} \sigma\left(W_1 \cdot F^k[v, :]\right)$

- **MAGNN** [23]:
  $G^t[v_j, :] = \sigma\left(W_2 \cdot \sum_{m \in \mathcal{M}} \beta_m \cdot \mathrm{Agg}_m(\mathrm{MP}_m(v_j))\right)$

## 5 Methodology

To address the class imbalance challenge in heterogeneous graphs, we propose the HetGSMOTE framework, which integrates representation learning, oversampling, adaptive edge generation, and classification into a unified pipeline. As illustrated in Fig. 2, our approach consists of four interconnected components: (1) a heterogeneous graph encoder for feature extraction, (2) SMOTE-based oversampling in the learned embedding space, (3) neural edge generators for synthetic node connectivity, and (4) a classifier for final predictions. The core components of HEtGSMOTE, the feature extraction, SMOTE oversampling and edge Generation has been illustrated in Fig. 1.

### 5.1 Heterogeneous Graph Encoder

The encoder serves as a feature extractor that leverages heterogeneous graph neural networks (HGNNs) such as HetGNN [5], MAGNN [23], and HAN [4] to generate meaningful node embeddings. We selected these HGNNs to cover a spectrum of mechanisms, including simple, attention-based, and metapath-based approaches. The encoder operates through two sequential aggregation mechanisms: content aggregation and type-specific neighbor aggregation, producing node representations that capture both attribute information and structural relationships within the heterogeneous graph.

#### 5.1.1 Content Aggregation

The content aggregation layer combines multiple attribute matrices associated with each node type. For a node $v_j$ of type $t$, we concatenate all available attribute matrices along the feature dimension and apply a linear transformation followed by a nonlinear activation to maintain the original embedding dimensionality:

$$F^t[v_j, :] = \sigma\left(W_c \cdot \left(\bigoplus_{i=1}^{n_{\mathrm{attr}}} F_i^t[v_j, :]\right)\right) \qquad (1)$$

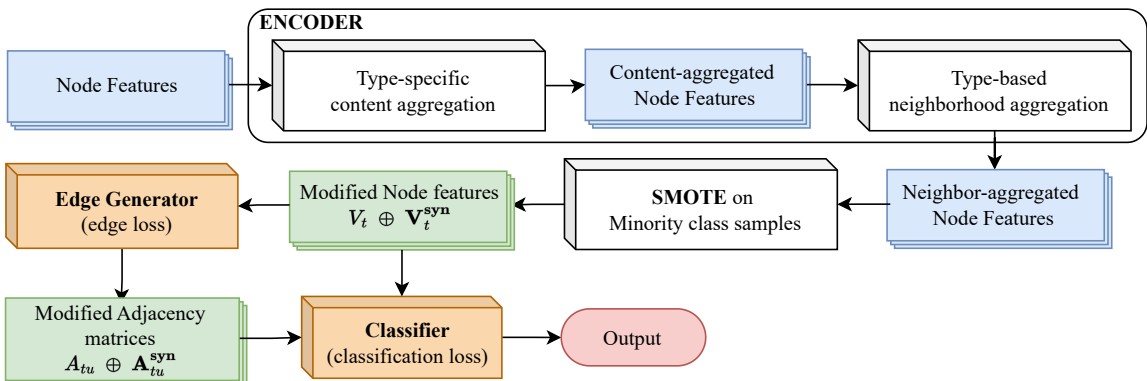

**Figure 2.** Illustration of the HetGSMOTE process, highlighting feature extraction, SMOTE-based node oversampling, neural edge generation, and classifier.

where $F_i^t \in \mathbb{R}^{n_t \times d_i}$ represents the $i$-th attribute matrix for nodes of type $t$, $n_{\text{attr}}$ is the total number of attribute matrices, $W_c \in \mathbb{R}^{d \times \sum_{i=1}^{n_{\text{attr}}} d_i}$ is the learnable weight matrix, $\bigoplus$ denotes concatenation, and $\sigma(\cdot)$ is the ReLU activation function.

The content-aggregated embeddings $F^t$ serve as input for subsequent type-specific neighbor aggregation and can also be used independently for baseline comparisons with vanilla SMOTE implementations.

### 5.1.2 Type-Specific Neighbor Aggregation

This component captures the heterogeneous structural information by aggregating neighbor representations across different node types. For each node, we first extract frequently occurring neighbors through random walks, selecting the top $m$ neighbors of each type to construct type-specific neighborhood representations.

The neighbor aggregation process varies depending on the chosen base HGNN architecture. Let $G^t[v_j, :]$ denote the aggregated embedding of type-$t$ neighbors for node $v_j$. The specific aggregation mechanisms for different base models were detailed in Section A.3 using (17), (18), and (19). For clarity and consistency, we reiterate the generalized formulation below:

$$G^t[v_j, :] = \text{BaseModel}\left(F^t[v_j, :], \mathcal{N}(v_j)\right)$$

After obtaining aggregated neighbor embeddings, we compute type-level attention weights to determine the importance of different node types:

$$\alpha_{v_j}^{t'} = \frac{\exp\left(\sigma\left(W_a \cdot (G^{t'}[v_j, :] \oplus F^t[v_j, :])\right)\right)}{\sum_{t' \in \mathcal{T}} \exp\left(\sigma\left(W_a \cdot (G^{t'}[v_j, :] \oplus F^t[v_j, :])\right)\right)} \quad (2)$$

and final node embedding combines the content-aggregated features with the weighted sum of type-specific neighbor embeddings as:

$$X^t[v_j, :] = W_f \cdot \left(F^t[v_j, :] \oplus \sum_{t' \in \mathcal{T}} \alpha_{v_j}^{t'} G^{t'}[v_j, :]\right) \quad (3)$$

where $W_a$ and $W_f$ are learnable weight matrices, and $\sigma(\cdot)$ represents the LeakyReLU activation function.

### 5.2 SMOTE-Based Oversampling

Following the embedding generation, we apply SMOTE to address class imbalance by generating synthetic minority class samples in the learned embedding space. This approach leverages the property that semantically similar nodes tend to cluster together in the embedding space after heterogeneous graph convolution.

For a randomly selected minority class node $v_j$ with embedding $X^t[v_j, :]$ and label $Y_{v_j}$, we identify its nearest neighbor from the same class:

$$\text{NN}(v_j) = \arg \min_{v \in V_t, Y_v = Y_{v_j}} \|X^t[v_j, :] - X^t[v, :]\|_2 \quad (4)$$

The synthetic node embedding is generated through linear interpolation between the source node and its nearest neighbor:

$$X_{\text{syn}}^t[v_j, :] = X^t[v_j, :] + \lambda \cdot (X^t[\text{NN}(v_j), :] - X^t[v_j, :]) \quad (5)$$

where $\lambda \sim \text{Uniform}(0, 1)$ is a random interpolation factor, and the synthetic node inherits the same label as the source nodes: $Y_{\text{syn}}[v_j] = Y_{v_j}$.

This process continues until the desired level of class balance is achieved, with all synthetic nodes forming the set $V_t^{\text{syn}}$. The key advantage of performing oversampling in the learned embedding space is that the generated samples are less prone to noise, as nodes of the same class naturally cluster together after the heterogeneous graph encoding process.

### 5.3 Neural Edge Generation

Since synthetic nodes are initially isolated from the graph structure, we employ neural edge generators to predict realistic connectivity patterns. The edge

generator is trained to reconstruct adjacency matrices for real nodes using their learned representations, enabling effective edge prediction for synthetic nodes.

For edge types between node types $t$ and $u$, the edge generator employs a bilinear transformation:

$$\hat{A}_{tu} = \sigma(X^t W_e X^{u\top}) \odot A_{tu} \qquad (6)$$

where $W_e \in \mathbb{R}^{d \times d}$ is a learnable weight matrix, $\sigma(\cdot)$ is the sigmoid activation function, and $\odot$ denotes element-wise multiplication with the original adjacency matrix $A_{tu}$ to focus learning on existing edge patterns.

The edge generator is trained using the reconstruction loss:

$$\mathcal{L}_{\text{edge}} = \sum_{(t,u) \in \mathcal{T} \times \mathcal{T}} \|\hat{A}_{tu}^{\text{real}} - A_{tu}^{\text{real}}\|_F^2 \qquad (7)$$

where $\hat{A}_{tu}^{\text{real}}$ represents the predicted adjacency matrix for real nodes, and $\|\cdot\|_F$ denotes the Frobenius norm.

We implement two strategies for incorporating predicted edges into the augmented graph:

**Hard Edge Strategy:** Binary edges are created using a threshold $\eta = 0.5$:

$$(A_{tu}^{\text{syn}})_{ij} = \begin{cases} 1, & \text{if } (\hat{A}_{tu})_{ij} \geq \eta \\ 0, & \text{if } (\hat{A}_{tu})_{ij} < \eta \end{cases} \qquad (8)$$

**Soft Edge Strategy:** Continuous edge weights are preserved to enable gradient flow:

$$(A_{tu}^{\text{syn}})_{ij} = (\hat{A}_{tu})_{ij} \quad \forall v_i \in V_t^{\text{syn}}, v_j \in V_u \qquad (9)$$

The soft edge strategy allows joint optimization of both edge prediction and node classification objectives, potentially leading to better overall performance.

## 5.4 Node Classification

The final component employs a heterogeneous GNN classifier that operates on the augmented graph containing both real and synthetic nodes with their predicted edges. The classifier utilizes a similar neighbor aggregation mechanism as the encoder to generate updated node embeddings that incorporate the relational information from predicted synthetic edges.

The classification process computes node predictions through an MLP head:

$$P_{v_j} = \text{softmax}(\text{MLP}(X'[v_j, :])) \qquad (10)$$

where $X'[v_j, :]$ represents the updated embedding for node $v_j$ after aggregating information from the augmented graph structure.

The classifier is trained using cross-entropy loss over all labeled nodes (real and synthetic):

$$\mathcal{L}_{\text{cls}} = - \sum_{v_j \in V_{\text{labeled}}} \sum_{c=1}^{C} \mathbb{I}[Y_{v_j} = c] \log P_{v_j}^{(c)} \qquad (11)$$

where $C$ is the number of classes, $\mathbb{I}[\cdot]$ is the indicator function, and $P_{v_j}^{(c)}$ is the predicted probability for class $c$.

For inference, the predicted class is determined as:

$$\hat{Y}_{v_j} = \arg\max_c P_{v_j}^{(c)} \qquad (12)$$

The overall training objective combines the edge generation and classification losses when using the soft edge strategy as $\mathcal{L}_{\text{total}} = \mathcal{L}_{\text{cls}} + \lambda \mathcal{L}_{\text{edge}}$ where $\lambda$ is a hyperparameter controlling the relative importance of edge reconstruction.

# 6 Experiments

## 6.1 Optimization

The optimization objective for HetGSMOTE involves the optimization of weights for the feature extractor, edge generator, and classifier. The model is optimized on two loss functions: the edge loss ($L_{\text{edge}}$) from the edge generator and the classification loss ($L_{\text{cls}}$) from the classifier. They are combined as $L = \lambda \cdot L_{\text{edge}} + L_{\text{cls}}$, where $\lambda$ is a hyperparameter that defines the relative importance of the two tasks. The resulting objective function of HetGSMOTE is the same as for GraphSMOTE:

$$\min_{\theta, \phi, \varphi} (L_{\text{cls}} + \lambda \cdot L_{\text{edge}}) \qquad (13)$$

where $\theta$, $\phi$, and $\varphi$ are the feature extractor, edge generator, and node classifier parameters, respectively. The pretraining of the feature extractor and edge generator using $L_{\text{edge}}$ was also explored to enhance the stability of the training.

## 6.2 Datasets

We procured diverse heterogeneous datasets of various sizes to show the domain independence of our method. These include AMiner (A-II) [5] and DBLP bibliographic datasets [23], movie collaboration-based IMDb [4], and biomedical PubMed [6] datasets. The summary and statistics of these datasets are included in the Appendix Section B and Table B.1.

## 6.3 Experimental Settings

To evaluate performance, we conduct node classification experiments with HetGSMOTE and baseline oversampling methods under various settings. Our focus in this work is on SMOTE-based oversampling

**Table 1.** Overview of Baselines and Experimental Settings

| Method | Code | Description |
|---|---|---|
| No oversampling | `no` | Original datasets without oversampling |
| Upsampling | `up` | Upsampling by duplicating source nodes with adjacency: $A_{tu}[v_j,:] = A_{tu}^0[v,:]$ |
| SMOTE[22] | `smote` | SMOTE on raw embedding space post-content aggregation with adjacency: $A_{tu}[v_j,:] = \min(1, A_{tu}^0[v_1,:] + A_{tu}^0[v_2,:])$ |
| Re-weight [16] | `reweight` | Cost-sensitive approach with higher loss weights for minority classes |
| Embed-SMOTE [24] | `embed_sm` | SMOTE after neighbor aggregation without edge generator, assuming sufficient relational information transfer |
| SMOTE + inherited edges | `em_smote` | SMOTE after neighbor aggregation with inherited adjacency matrix to demonstrate edge generator impact |
| **HetGSMOTE (Ours)** | `HetGSM` | Framework with edge generator and encoder pre-trained on edge prediction, fine-tuned on edge loss and classification loss |

**Table 2.** Test Accuracy of HetGSMOTE (with HGNN) vs Upsampling Ratios for IMDb datasets

| Setting | 0.2 | 0.4 | 0.6 | 0.8 | 1.0 | 1.2 |
|---|---|---|---|---|---|---|
| no | $0.505 \pm 0.027$ | $0.496 \pm 0.023$ | $0.505 \pm 0.025$ | $0.501 \pm 0.021$ | $0.505 \pm 0.027$ | $0.509 \pm 0.021$ |
| up | $0.500 \pm 0.029$ | $0.501 \pm 0.026$ | $0.505 \pm 0.024$ | $0.504 \pm 0.027$ | $0.499 \pm 0.021$ | $0.509 \pm 0.024$ |
| smote | $\mathbf{0.507 \pm 0.019}$ | $0.500 \pm 0.015$ | $0.504 \pm 0.018$ | $0.498 \pm 0.019$ | $0.502 \pm 0.020$ | $0.505 \pm 0.023$ |
| reweight | $0.500 \pm 0.019$ | $0.501 \pm 0.013$ | $0.504 \pm 0.022$ | $0.503 \pm 0.019$ | $0.500 \pm 0.026$ | $0.500 \pm 0.018$ |
| embed_sm | $0.503 \pm 0.020$ | $0.500 \pm 0.020$ | $0.496 \pm 0.020$ | $0.506 \pm 0.019$ | $0.502 \pm 0.018$ | $0.496 \pm 0.018$ |
| em_smote | $0.495 \pm 0.021$ | $\mathbf{0.508 \pm 0.025}$ | $0.498 \pm 0.026$ | $0.502 \pm 0.020$ | $0.503 \pm 0.027$ | $0.494 \pm 0.021$ |
| **HetGSM** | $0.504 \pm 0.019$ | $0.507 \pm 0.020$ | $\mathbf{0.506 \pm 0.018}$ | $\mathbf{0.508 \pm 0.019}$ | $\mathbf{0.509 \pm 0.021}$ | $\mathbf{0.515 \pm 0.022}$ |

methods, chosen for their simplicity, interpretability, and lower computational overhead compared to adversarial or autoencoder-based approaches. The baselines are listed in Table 1.

For the HetGSMOTE (HetGSM) method , we will pre-train the encoder and edge generator on the edge prediction task since the model can take advantage of prior knowledge of general graph structure. We will also be using soft synthetic edges, as mentioned in (9), for the framework. As a result, the encoder and edge generator will be fine-tuned on both edge loss and classification loss.

### 6.4 Evaluation Metrics

To comprehensively evaluate the performance of our proposed HetGSMOTE method on imbalanced heterogeneous graph node classification, we employ three widely-adopted evaluation metrics following the experimental setup of GraphSMOTE: macro-averaged accuracy (ACC), macro-averaged AUC-ROC score (AUC), and macro-averaged F1-score (F1). The macro-averaging approach for evaluation metrics ensures that performance on minority classes receives equal weight to majority classes, providing a more balanced assessment of model effectiveness in imbalanced scenarios. The details of the evaluation metrics and formulas are given in the Appendix Section C.

### 6.5 Experimental Configuration

Experiments run on NVIDIA A100 80GB GPU using ADAM optimizer (lr=1e-4, weight decay=5e-4). Regularization $\lambda$: 1e-6 for HetGSMOTE with synthetic soft edges, 10 otherwise. Models trained for

200 epochs with embedding dimensions 128-256. Results averaged over 10 independent runs with fixed seeds across different imbalance ratios and upsampling ratios, with synthetic edges incorporated only during training.

## 7 Results

Results span all base models, datasets, and imbalance ratios using 3 metrics. We present the most significant findings for each evaluation context. All values represent means across 10 training subsets. Bold indicates best performance within each imbalance/upsampling comparison. We also include the extended results with F1 and AUC tables in GitHub and Appendix D.

### 7.1 Influence of Up-sampling Ratio

The *upsampling ratio* defines the fractional increase in minority samples to overcome the imbalance. An upsampling ratio of 1 implies the minority size was doubled by oversampling. Experiments with the upsampling ratio highlight the learning curve of our oversampling technique in different oversampling stages. In the experiments, the upsampling ratio scale is varied in the range $[0.2, 1.2]$ in 0.2 steps to obtain results for both high and low ratio cases while keeping the imbalance ratio to 0.5. The results were obtained from HetGMOTE with HGNN base model on IMDb dataset and are tabulated in Table 2.

We find that HetGSMOTE outperforms all the baselines regularly from the 0.6 ratio threshold, as seen in the table. This shows that the generated node samples have an impact on the performance

without fail. For lower ratios, smote, and em_smote gave better results. Baseline em_smote performing better might indicate the less importance of the edge generators when synthetic samples are low, which can affect the classification negatively compared to simpler oversampling.

## 7.2 Performance of HetGSMOTE

To show the superior performance of the Het-GSMOTE framework, we show its performance with the HAN base model on IMDb datasets across all metrics. With the IMDb dataset being small with a fairly balanced distribution of node types and HAN the best-performing model we have, we substantiate the choice of experiment showcased here. The results are tabulated in Table 3. HetGSMOTE has shown higher performance than its counterparts in most cases across all three metrics, especially in the severe imbalance (low IR) scenarios.

## 7.3 Influence of Base Model

We evaluate HetGSMOTE with HAN, MAGNN, and HGNN base models using test accuracy on the IMDb dataset (Table 4). HAN achieves the best performance among base models, reflecting different model capacities for learning heterogeneous graph structures. Importantly, HetGSMOTE consistently outperforms baselines across all base models, demonstrating the framework's effectiveness regardless of the underlying architecture.

## 7.4 Variation across datasets

To show the consistency of performance across different datasets, we have shown the results in Table 5 with the test accuracy performance of HetGSMOTE on all datasets. We note that that Heterogeneous graph datasets with imbalanced node-type distributions often show greater performance fluctuations, as class imbalance interacts with structural imbalance.

For clarity and focus, the comparison is limited to the most competitive baseline methods. Here, we make the following observation: (1) the Het-GSMOTE performs comparatively better than the baselines in most cases for all datasets. (2) Significant performance gains are observed under conditions of severe class imbalance.

## 7.5 Ablation Studies

We conduct ablation experiments to evaluate the individual contributions of key components in the HetGSMOTE framework discussed in Section 6.3. We systematically examine variants by selectively including or excluding specific framework elements. The HetGSMOTE framework incorporates two primary components: pre-training for edge prediction tasks and soft synthetic edges. We evaluate four distinct variants: GSM_base (neither component), GSM_S (soft edges only), GSM_PT (pre-training only), and HetGSM (both components combined).

The soft synthetic edges implementation trains both the encoder and edge generator using classification loss combined with edge loss, as detailed in (9). Pre-training focuses exclusively on the edge prediction task.

### 7.5.1 Imbalance Ratio Analysis

We compare these variants using the HAN base model on the IMDb dataset under low imbalance ratio conditions, consistent with the rationale provided in Section 7.2. Table 6 presents the comparative performance results. Similar trends have been observed for other datasets as well. The complete HetGSM variant incorporating both pre-training and synthetic soft edges demonstrates superior performance across most imbalance ratios, achieving the highest scores in four out of five test conditions.

### 7.5.2 Upsampling Ratio Analysis

We examine overgeneration through upsampling ratio ablation studies. Table 7 reveals optimal upsampling thresholds. Results show two findings: First, above the 0.6 threshold from Section 7.1, HetGSM consistently outperforms alternatives. Second, three approaches degrade when transitioning from ratio 1.0 to 1.2, indicating excessive synthetic generation compromises embedding quality.

## 8 Conclusion

Our findings demonstrate the effectiveness of Het-GSMOTE for addressing class imbalance in heterogeneous graphs across various datasets, base models, and imbalance conditions. The approach is domain-independent, consistently surpassing baselines across diverse datasets including AMiner, DBLP, IMDb, and PubMed. The framework's flexibility allows it to work with different base models (HetGNN, HAN, MAGNN) and shows promise for more.

Key insights from our experiments include: (1) performance boost is significant under severe imbalance, (2) very low upsampling ratios hinder effectiveness, and (3) pretraining the feature extractor and edge generator significantly boosts performance.

Future work will extend the approach to additional tasks including link prediction, edge type classification, and node representation learning. Additionally methods such as BARE and FinC-GAN, compatible with our framework, can be investigated with HetGSMOTE to enhance robustness. Future directions could explore integrating our approach with privacy-preserving techniques [25].

**Table 3.** Performance of baselines and HetGSMOTE (HAN base model) vs. imbalance ratio for IMDb dataset

| # | Settings | 0.1 | 0.2 | 0.3 | 0.4 | 0.5 | 0.6 | 0.7 | 0.8 | 0.9 |
|---|---|---|---|---|---|---|---|---|---|---|
| Accuracy | no | 0.447 ± 0.018 | 0.458 ± 0.021 | 0.479 ± 0.023 | 0.500 ± 0.025 | 0.517 ± 0.037 | 0.515 ± 0.026 | 0.525 ± 0.036 | 0.516 ± 0.029 | 0.518 ± 0.029 |
| | up | 0.443 ± 0.023 | 0.456 ± 0.026 | 0.472 ± 0.024 | 0.485 ± 0.028 | 0.511 ± 0.024 | 0.508 ± 0.029 | 0.520 ± 0.031 | 0.520 ± 0.020 | 0.523 ± 0.024 |
| | smote | 0.451 ± 0.022 | 0.456 ± 0.030 | 0.477 ± 0.029 | 0.495 ± 0.023 | 0.509 ± 0.032 | 0.525 ± 0.025 | 0.521 ± 0.028 | 0.517 ± 0.030 | 0.521 ± 0.032 |
| | reweight | 0.452 ± 0.015 | 0.476 ± 0.024 | **0.499 ± 0.032** | 0.518 ± 0.020 | 0.515 ± 0.019 | 0.523 ± 0.017 | 0.528 ± 0.019 | 0.531 ± 0.021 | **0.534 ± 0.029** |
| | embed_sm | 0.445 ± 0.017 | 0.450 ± 0.029 | 0.460 ± 0.024 | 0.481 ± 0.029 | 0.490 ± 0.028 | 0.499 ± 0.027 | 0.507 ± 0.026 | 0.509 ± 0.026 | 0.519 ± 0.024 |
| | em_smote | 0.455 ± 0.016 | 0.487 ± 0.028 | 0.494 ± 0.036 | 0.503 ± 0.024 | 0.517 ± 0.025 | 0.524 ± 0.028 | 0.529 ± 0.026 | 0.527 ± 0.025 | 0.529 ± 0.021 |
| | **HetGSM** | **0.472 ± 0.014** | **0.489 ± 0.017** | 0.495 ± 0.024 | **0.518 ± 0.015** | **0.528 ± 0.035** | **0.537 ± 0.017** | 0.531 ± 0.023 | **0.544 ± 0.026** | 0.534 ± 0.031 |
| F1-Score | no | 0.550 ± 0.018 | 0.554 ± 0.012 | 0.557 ± 0.016 | 0.564 ± 0.013 | **0.573 ± 0.016** | 0.563 ± 0.009 | 0.570 ± 0.015 | 0.566 ± 0.017 | 0.568 ± 0.018 |
| | up | 0.545 ± 0.013 | 0.546 ± 0.017 | 0.551 ± 0.010 | 0.553 ± 0.015 | 0.559 ± 0.015 | 0.562 ± 0.017 | 0.567 ± 0.020 | 0.561 ± 0.014 | 0.566 ± 0.019 |
| | smote | 0.545 ± 0.014 | 0.550 ± 0.016 | 0.549 ± 0.018 | 0.558 ± 0.011 | 0.561 ± 0.020 | 0.568 ± 0.013 | 0.567 ± 0.017 | 0.566 ± 0.017 | 0.565 ± 0.014 |
| | reweight | 0.541 ± 0.011 | 0.551 ± 0.016 | 0.556 ± 0.016 | 0.560 ± 0.012 | 0.569 ± 0.010 | 0.567 ± 0.018 | 0.569 ± 0.014 | 0.574 ± 0.010 | **0.581 ± 0.017** |
| | embed_sm | 0.544 ± 0.010 | 0.543 ± 0.019 | 0.545 ± 0.012 | 0.549 ± 0.011 | 0.555 ± 0.014 | 0.555 ± 0.007 | 0.560 ± 0.017 | 0.558 ± 0.014 | 0.571 ± 0.012 |
| | em_smote | 0.547 ± 0.021 | 0.553 ± 0.015 | 0.556 ± 0.016 | 0.558 ± 0.012 | 0.568 ± 0.014 | 0.568 ± 0.012 | 0.572 ± 0.019 | 0.566 ± 0.018 | 0.568 ± 0.013 |
| | **HetGSM** | **0.553 ± 0.010** | **0.561 ± 0.017** | **0.562 ± 0.016** | **0.571 ± 0.021** | 0.570 ± 0.024* | **0.573 ± 0.015** | **0.575 ± 0.017** | **0.576 ± 0.019** | 0.579 ± 0.018 |
| AUC | no | 0.661 ± 0.030 | 0.670 ± 0.022 | 0.682 ± 0.028 | 0.699 ± 0.018 | **0.706 ± 0.021** | 0.699 ± 0.014 | 0.707 ± 0.017 | 0.703 ± 0.022 | 0.707 ± 0.020 |
| | up | 0.652 ± 0.023 | 0.662 ± 0.029 | 0.669 ± 0.020 | 0.675 ± 0.023 | 0.685 ± 0.018 | 0.692 ± 0.021 | 0.702 ± 0.028 | 0.694 ± 0.020 | 0.701 ± 0.024 |
| | smote | 0.658 ± 0.030 | 0.669 ± 0.027 | 0.670 ± 0.028 | 0.685 ± 0.011 | 0.692 ± 0.028 | 0.703 ± 0.020 | 0.699 ± 0.022 | 0.704 ± 0.023 | 0.703 ± 0.023 |
| | reweight | 0.645 ± 0.025 | 0.679 ± 0.029 | 0.686 ± 0.021 | 0.693 ± 0.016 | 0.703 ± 0.019 | 0.700 ± 0.020 | 0.703 ± 0.021 | 0.707 ± 0.018 | 0.716 ± 0.024 |
| | embed_sm | 0.646 ± 0.018 | 0.651 ± 0.031 | 0.660 ± 0.025 | 0.670 ± 0.018 | 0.676 ± 0.024 | 0.683 ± 0.017 | 0.689 ± 0.023 | 0.691 ± 0.020 | 0.705 ± 0.019 |
| | em_smote | 0.665 ± 0.031 | 0.680 ± 0.028 | 0.690 ± 0.021 | 0.690 ± 0.018 | 0.701 ± 0.018 | 0.704 ± 0.020 | **0.712 ± 0.021** | 0.702 ± 0.023 | 0.706 ± 0.014 |
| | **HetGSM** | **0.667 ± 0.020** | **0.685 ± 0.024** | **0.692 ± 0.019** | **0.705 ± 0.022** | 0.702 ± 0.023* | **0.713 ± 0.016** | 0.705 ± 0.020 | **0.713 ± 0.023** | **0.717 ± 0.024** |

*Accuracy is higher for our method; the other metrics are comparable within run-to-run variance.

**Table 4.** Test Accuracy for HetGSMOTE (all base models) vs imbalance ratio for IMDb dataset

| # | Settings | 0.1 | 0.2 | 0.3 | 0.4 | 0.5 | 0.6 | 0.7 | 0.8 | 0.9 |
|---|---|---|---|---|---|---|---|---|---|---|
| HAN | reweight | 0.452 ± 0.015 | 0.476 ± 0.024 | **0.499 ± 0.032** | 0.518 ± 0.020 | 0.515 ± 0.019 | 0.523 ± 0.017 | 0.528 ± 0.019 | 0.531 ± 0.021 | **0.534 ± 0.029** |
| | embed_sm | 0.445 ± 0.017 | 0.450 ± 0.029 | 0.460 ± 0.024 | 0.481 ± 0.029 | 0.490 ± 0.028 | 0.499 ± 0.027 | 0.507 ± 0.026 | 0.509 ± 0.026 | 0.519 ± 0.024 |
| | em_smote | 0.455 ± 0.016 | 0.487 ± 0.028 | 0.494 ± 0.036 | 0.503 ± 0.024 | 0.517 ± 0.025 | 0.524 ± 0.028 | 0.529 ± 0.026 | 0.527 ± 0.025 | 0.529 ± 0.021 |
| | **HetGSM** | **0.472 ± 0.014** | **0.489 ± 0.017** | 0.495 ± 0.024 | **0.518 ± 0.015** | **0.528 ± 0.035** | **0.537 ± 0.017** | 0.531 ± 0.023 | **0.544 ± 0.026** | 0.534 ± 0.031 |
| MAGNN | reweight | 0.357 ± 0.022 | 0.348 ± 0.022 | 0.367 ± 0.039 | 0.406 ± 0.067 | 0.338 ± 0.011 | 0.352 ± 0.038 | 0.348 ± 0.024 | 0.338 ± 0.008 | 0.340 ± 0.008 |
| | embed_sm | 0.346 ± 0.011 | 0.346 ± 0.023 | 0.348 ± 0.025 | 0.350 ± 0.025 | 0.346 ± 0.015 | 0.339 ± 0.008 | 0.345 ± 0.012 | 0.342 ± 0.013 | 0.363 ± 0.023 |
| | em_smote | 0.430 ± 0.044 | 0.470 ± 0.019 | 0.488 ± 0.024 | **0.501 ± 0.021** | 0.474 ± 0.045 | 0.497 ± 0.031 | **0.513 ± 0.018** | 0.489 ± 0.034 | 0.504 ± 0.031 |
| | **HetGSM** | **0.454 ± 0.026** | **0.475 ± 0.016** | **0.489 ± 0.023** | 0.498 ± 0.026 | **0.510 ± 0.026** | **0.525 ± 0.022** | 0.512 ± 0.027 | **0.520 ± 0.027** | **0.509 ± 0.024** |
| HGNN | reweight | 0.434 ± 0.016 | 0.440 ± 0.033 | 0.456 ± 0.028 | 0.466 ± 0.028 | 0.489 ± 0.028 | 0.488 ± 0.029 | **0.508 ± 0.041** | 0.499 ± 0.021 | **0.512 ± 0.030** |
| | embed_sm | 0.435 ± 0.019 | **0.457 ± 0.024** | 0.466 ± 0.016 | 0.475 ± 0.035 | 0.500 ± 0.028 | 0.494 ± 0.035 | 0.503 ± 0.025 | 0.501 ± 0.019 | 0.495 ± 0.022 |
| | em_smote | 0.434 ± 0.022 | 0.442 ± 0.023 | 0.456 ± 0.027 | 0.479 ± 0.026 | 0.494 ± 0.031 | 0.501 ± 0.025 | 0.497 ± 0.031 | 0.498 ± 0.028 | 0.501 ± 0.031 |
| | **HetGSM** | **0.444 ± 0.018** | 0.448 ± 0.021 | **0.476 ± 0.023** | **0.495 ± 0.025** | **0.503 ± 0.033** | **0.508 ± 0.024** | 0.498 ± 0.024 | **0.507 ± 0.022** | 0.508 ± 0.021 |

**Table 5.** Test Accuracy of HetGSMOTE (HAN base model) vs imbalance ratios for all datasets

| # | Settings | 0.1 | 0.2 | 0.3 | 0.4 | 0.5 | 0.6 | 0.7 | 0.8 | 0.9 |
|---|---|---|---|---|---|---|---|---|---|---|
| IMDb | reweight | 0.452 ± 0.015 | 0.476 ± 0.024 | **0.499 ± 0.032** | 0.518 ± 0.020 | 0.515 ± 0.019 | 0.523 ± 0.017 | 0.528 ± 0.019 | 0.531 ± 0.021 | **0.534 ± 0.029** |
| | embed_sm | 0.445 ± 0.017 | 0.450 ± 0.029 | 0.460 ± 0.024 | 0.481 ± 0.029 | 0.490 ± 0.028 | 0.499 ± 0.027 | 0.507 ± 0.026 | 0.509 ± 0.026 | 0.519 ± 0.024 |
| | em_smote | 0.455 ± 0.016 | 0.487 ± 0.028 | 0.494 ± 0.036 | 0.503 ± 0.024 | 0.517 ± 0.025 | 0.524 ± 0.028 | 0.529 ± 0.026 | 0.527 ± 0.025 | 0.529 ± 0.021 |
| | **HetGSM** | **0.472 ± 0.014** | **0.489 ± 0.017** | 0.495 ± 0.024 | **0.518 ± 0.015** | **0.528 ± 0.035** | **0.537 ± 0.017** | 0.531 ± 0.023 | **0.544 ± 0.026** | 0.534 ± 0.031 |
| AMiner | reweight | 0.922 ± 0.020 | 0.949 ± 0.011 | 0.957 ± 0.011 | 0.959 ± 0.010 | 0.964 ± 0.007 | 0.962 ± 0.010 | 0.965 ± 0.008 | 0.963 ± 0.012 | 0.961 ± 0.009 |
| | embed_sm | 0.932 ± 0.022 | 0.952 ± 0.013 | 0.962 ± 0.012 | 0.963 ± 0.011 | 0.958 ± 0.013 | **0.967 ± 0.010** | 0.960 ± 0.013 | **0.965 ± 0.008** | 0.963 ± 0.013 |
| | em_smote | 0.948 ± 0.022 | 0.956 ± 0.015 | 0.964 ± 0.013 | 0.962 ± 0.009 | **0.965 ± 0.012** | 0.962 ± 0.012 | 0.967 ± 0.012 | 0.964 ± 0.009 | 0.963 ± 0.014 |
| | **HetGSM** | **0.951 ± 0.012** | **0.961 ± 0.010** | **0.965 ± 0.013** | **0.966 ± 0.013** | 0.964 ± 0.012 | 0.964 ± 0.012 | **0.967 ± 0.014** | 0.962 ± 0.015 | **0.967 ± 0.013** |
| PubMed | reweight | 0.168 ± 0.019 | 0.168 ± 0.019 | 0.153 ± 0.013 | **0.154 ± 0.015** | 0.158 ± 0.020 | 0.158 ± 0.017 | **0.164 ± 0.016** | 0.156 ± 0.016 | 0.153 ± 0.009 |
| | embed_sm | 0.140 ± 0.015 | 0.149 ± 0.015 | 0.164 ± 0.011 | 0.151 ± 0.012 | 0.158 ± 0.006 | 0.153 ± 0.012 | 0.154 ± 0.015 | 0.153 ± 0.016 | **0.159 ± 0.019** |
| | em_smote | 0.154 ± 0.021 | 0.154 ± 0.021 | 0.157 ± 0.016 | 0.153 ± 0.013 | 0.154 ± 0.013 | **0.160 ± 0.012** | 0.156 ± 0.017 | 0.146 ± 0.015 | 0.154 ± 0.018 |
| | **HetGSM** | **0.170 ± 0.018** | **0.171 ± 0.017** | **0.169 ± 0.016** | 0.151 ± 0.012 | **0.163 ± 0.019** | 0.150 ± 0.016 | 0.158 ± 0.017 | 0.156 ± 0.017 | 0.157 ± 0.013 |
| DBLP | reweight | 0.646 ± 0.021 | 0.690 ± 0.044 | 0.735 ± 0.049 | **0.770 ± 0.055** | **0.800 ± 0.036** | **0.805 ± 0.018** | **0.776 ± 0.046** | **0.791 ± 0.048** | 0.777 ± 0.046 |
| | embed_sm | 0.637 ± 0.024 | 0.683 ± 0.040 | 0.705 ± 0.039 | 0.738 ± 0.043 | 0.767 ± 0.023 | 0.769 ± 0.030 | 0.768 ± 0.033 | 0.774 ± 0.030 | 0.781 ± 0.036 |
| | em_smote | 0.645 ± 0.039 | 0.684 ± 0.049 | 0.715 ± 0.048 | 0.746 ± 0.044 | 0.764 ± 0.040 | 0.775 ± 0.022 | 0.736 ± 0.051 | 0.762 ± 0.034 | 0.751 ± 0.059 |
| | **HetGSM** | **0.665 ± 0.032** | **0.714 ± 0.055** | **0.744 ± 0.043** | 0.762 ± 0.048 | 0.761 ± 0.040 | 0.767 ± 0.030 | 0.764 ± 0.045 | 0.784 ± 0.031 | **0.787 ± 0.044** |

**Table 6.** Test Accuracy of HetGSMOTE variants (HAN base model) vs. imbalance ratio for IMDb dataset

| Setting | 0.1 | 0.2 | 0.3 | 0.4 | 0.5 |
|---|---|---|---|---|---|
| GSM_base | 0.449 ± 0.018 | 0.481 ± 0.022 | 0.495 ± 0.032 | 0.501 ± 0.028 | 0.520 ± 0.032 |
| GSM_S | 0.458 ± 0.026 | 0.483 ± 0.031 | 0.488 ± 0.029 | 0.505 ± 0.032 | 0.524 ± 0.029 |
| GSM_PT | 0.471 ± 0.013 | 0.484 ± 0.026 | **0.505 ± 0.027** | 0.516 ± 0.015 | 0.520 ± 0.025 |
| **HetGSM** | **0.472 ± 0.014** | **0.489 ± 0.017** | 0.495 ± 0.024 | **0.518 ± 0.015** | **0.528 ± 0.035** |

**Table 7.** Test Accuracy of HetGSMOTE variants (HGNN) vs. Upsampling Ratio for IMDb dataset

| Setting | 0.6 | 0.8 | 1.0 | 1.2 |
|---|---|---|---|---|
| GSM_base | 0.493 ± 0.018 | 0.494 ± 0.023 | 0.501 ± 0.027 | 0.492 ± 0.012 |
| GSM_S | 0.501 ± 0.011 | 0.501 ± 0.021 | 0.495 ± 0.019 | 0.490 ± 0.024 |
| GSM_PT | 0.506 ± 0.024 | 0.503 ± 0.016 | **0.510 ± 0.022** | 0.506 ± 0.022 |
| **HetGSM** | **0.506 ± 0.018** | **0.508 ± 0.019** | 0.509 ± 0.021 | **0.515 ± 0.022** |

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

# A   Background - Detailed

## A.1   Class Imbalance

Given a dataset with $k$ classes represented as $\{c_1, \ldots, c_k\}$, let $|c_i|$ denote the number of samples in class $c_i$. To quantify the degree of class imbalance, we employ the imbalance ratio (IR) for each class $i$:

$$\text{IR}_i = \frac{|c_i|}{\max_{j \in \{1,\ldots,k\}}(|c_j|)} \tag{14}$$

where $\max_j |c_j|$ represents the size of the largest class. A perfectly balanced dataset has $\text{IR}_i = 1$ for all classes, while lower values of $\text{IR}_i$ indicate higher imbalance, with minority classes having significantly fewer samples than the majority class.

## A.2   SMOTE

Synthetic Minority Oversampling Technique (SMOTE) [22] addresses class imbalance by generating synthetic samples for minority classes through interpolation rather than simple duplication. Given two feature vectors $X_1$ and $X_2$ from the same minority class, SMOTE creates synthetic samples using linear interpolation:

$$\text{SMOTE}(X_1, X_2) = (1 - r)X_1 + rX_2 \tag{15}$$

where $r \sim \text{Uniform}(0, 1)$ is a random interpolation parameter. This approach generates synthetic samples along the line segment connecting existing minority samples, effectively expanding the decision boundary for minority classes.

## A.3 Heterogeneous Graph Neural Networks

Heterogeneous graphs (HetGs) extend traditional homogeneous graphs by incorporating multiple node types and edge types, enabling richer semantic relationship modeling in domains such as bibliographic networks and knowledge graphs. HetG learning has evolved from manual feature engineering to sophisticated representation learning approaches, broadly categorized into shallow and deep models [26].

Shallow models include random walk-based methods like metapath2vec, which employs metapath-guided random walks, and matrix decomposition approaches such as HERec [2]. DeepWalk introduced SkipGram embeddings for capturing node co-occurrence probabilities, with variants like Spacey, JUST, and HHNE [3] incorporating enhancements such as jump-stay strategies and heterogeneous random walks.

Deep models leverage Heterogeneous Graph Neural Networks (HGNNs), including unsupervised methods like HetGNN [5] and semi-supervised approaches like HAN [4] that utilize attention mechanisms. Advanced HGNNs such as MAGNN and HGT [23] further refine intra- and inter-metapath aggregations. Additional techniques include encoder-decoder models [3] and adversarial frameworks like GraphGAN [27].

In this work, we employ three representative base models: HetGNN (HGNN), HAN, and MAGNN. Each base model updates node features by aggregating information from neighbors through distinct mechanisms. The general form for updating node $v_j$ of type $t$ with initial features $F^t[v_j, :]$ can be expressed as:

$$G^t[v_j, :] = \text{BaseModel}\left(F^t[v_j, :], \mathcal{N}(v_j)\right) \quad (16)$$

where $G^t[v_j, :]$ represents the aggregated node features and $\mathcal{N}(v_j)$ denotes the neighborhood of node $v_j$. The specific aggregation mechanisms are:

### A.3.1 HetGNN (HGNN)

Performs mean aggregation over transformed neighbor features:

$$G^t[v_j, :] = \frac{1}{|\mathcal{N}^t(v_j)|} \sum_{v \in \mathcal{N}^t(v_j)} \sigma\left(W \cdot F^t[v, :]\right) \quad (17)$$

where $\mathcal{N}^t(v_j)$ denotes neighbors of node $v_j$ of type $t$, $W$ is a trainable weight matrix, and $\sigma$ represents the ReLU activation function.

### A.3.2 HAN

Employs type-specific attention mechanisms across heterogeneous neighbors:

$$G^t[v_j, :] = \sum_k \sum_{v \in \mathcal{N}^k(v_j)} \alpha_{v, v_j} \sigma\left(W_1 \cdot F^k[v, :]\right) \quad (18)$$

where $\mathcal{N}^k(v_j)$ represents neighbors of type $k$, $\alpha_{v, v_j}$ denotes the attention weight between nodes $v$ and $v_j$, $W_1$ is a learnable transformation matrix, and $\sigma$ is the LeakyReLU activation.

### A.3.3 MAGNN

Incorporates metapath-based context through hierarchical intra-metapath and inter-metapath aggregation:

$$G^t[v_j, :] = \sigma\left(W_2 \cdot \sum_{m \in \mathcal{M}} \beta_m \cdot \text{Agg}_m(\text{MP}_m(v_j))\right) \quad (19)$$

where $\mathcal{M}$ represents the set of metapaths, $\text{Agg}_m(\text{MP}_m(v_j))$ denotes intra-metapath aggregation for metapath $m$ containing node $v_j$, $\beta_m$ represents the attention weight for metapath $m$, and $W_2$ is a learnable weight matrix.

## B Datasets

THe datasets, as discussed in Section 6.2, are given below.

1. **AMiner-AII** [5]: This is an academic dataset that includes paper publications in top venues related to artificial intelligence and data science from year 2006 to 2015. Each paper has various bibliographic content information: title (128 dim) and abstract (128 dim). The author and venue attributes are extracted from the random walks in the graph using Par2Vec [28].

2. **DBLP**[23]: This is a subset of the DBLP computer science bibliography website dataset collected from [23] containing bibliographic information for four node types with corresponding binary attributes (256 dim). The representation words of their paper keywords describe the author node's features.

3. **IMDb** [4]: This is a subset of the Internet Movie Database (IMDb) dataset collected from [4]. This is a movie-based dataset where movies are divided into three classes (action, comedy, drama) according to their genre, and their features are derived from the representation words of the plot keywords (128 dim). The attributes of other nodes are determined from the random walks using Par2Vec [28].

**Table B.1.** Information about datasets (labeled node type marked with $^*$)

| Dataset | Nodes | | Edges | | Classes |
|---|---|---|---|---|---|
| | Type | Count | Type | Count | (Minority) |
| AMiner-AII | author$^*$ | 20,171 | author-paper | 42,379 | 4 (3) |
| | paper | 13,250 | paper-paper | 14,583 | |
| | venue | 18 | paper-venue | 13,250 | |
| IMDb | movie$^*$ | 4,666 | movie-actor | 13,990 | 3 (2) |
| | director | 2,271 | movie-director | 4,666 | |
| | actor | 5,845 | | | |
| DBLP | author | 4,057 | author-paper | 19,645 | 4 (3) |
| | paper$^*$ | 14,328 | paper-term | 85,810 | |
| | term | 7,723 | paper-conference | 14,328 | |
| | conference | 20 | | | |
| PubMed | 4 types | 63,109 | 10 types | 244,986 | 8 (6) |

4. **PubMed** [6]: This is bio-medical data describing relations between genes, diseases, chemicals, and species with corresponding attributes (200 dim). The links include gene-gene interactions, gene-disease associations, chemical-species relationships, etc. There are 8 target labels for the disease nodes.

To prepare the experimental datasets with varying imbalance ratios, we followed the procedure of GraphSMOTE [9]: majority classes were downsampled to match the second-largest class in each graph, after which minority class nodes were further downsampled to obtain the desired imbalance ratio. For the DBLP dataset, where the smallest class contains only 20 samples, we instead selected another minority class with sufficient samples, since the standard definition of the imbalance ratio (smallest-to-largest class) does not apply. For all datasets, the adjacency matrix was preserved for the selected nodes, ensuring consistency with the original graph structure.

## C    Evaluation Metrics

**Macro-averaged Accuracy (ACC)** computes the classification accuracy independently for each class and then averages across all classes:

$$\text{ACC}_{\text{macro}} = \frac{1}{|\mathcal{C}|} \sum c \in \mathcal{C} \text{ACC}_c \qquad (20)$$

where the accuracy for each class $c$ is defined as:

$$\text{ACC}_c = \frac{TP_c + TN_c}{TP_c + TN_c + FP_c + FN_c} \qquad (21)$$

**Macro-averaged AUC-ROC (AUC)** computes the area under the receiver operating characteristic curve for each class independently, then averages across all classes:

$$\text{AUC}_{\text{macro}} = \frac{1}{|\mathcal{C}|} \sum_{c \in \mathcal{C}} \text{AUC}_c \qquad (22)$$

where $\mathcal{C}$ represents the set of all node classes and $\text{AUC}_c$ denotes the AUC score for class $c$ computed in a one-versus-rest manner.

**Macro-averaged F1-score (F1)** calculates the harmonic mean of precision and recall for each class, then averages across all classes:

$$\text{F1}_{\text{macro}} = \frac{1}{|\mathcal{C}|} \sum_{c \in \mathcal{C}} \text{F1}_c \qquad (23)$$

where the F1-score for each class $c$ is defined as:

$$\text{F1}_c = \frac{2 \cdot \text{Precision}_c \cdot \text{Recall}_c}{\text{Precision}_c + \text{Recall}_c} \qquad (24)$$

with precision and recall computed as:

$$\text{Precision}_c = \frac{TP_c}{TP_c + FP_c}, \quad \text{Recall}_c = \frac{TP_c}{TP_c + FN_c} \qquad (25)$$

Here, $TP_c$, $TN_c$, $FP_c$, and $FN_c$ denote the true positives, true negatives, false positives, and false negatives for class $c$, respectively. For fair comparison, the classification threshold for F1-score computation is optimally tuned to maximize performance across all baseline methods.

## D    Extended Results

This section presents extended evaluation results to verify that the trends reported in the main text remain consistent across metrics beyond test accuracy. In particular, we include additional F1 and AUC tables, which reinforce that the relative performance differences across models, datasets, and imbalance ratios follow the same patterns, with only minor numerical variations.

### D.1    Influence of Base Model

Tables D.1 and D.2 report the F1 and AUC scores for HAN, MAGNN, and HGNN on the IMDb dataset,

**Table D.1.** F1 scores for HetGSMOTE (across all base models) under varying imbalance ratios on the IMDb dataset

| # | Settings | 0.1 | 0.2 | 0.3 | 0.4 | 0.5 | 0.6 | 0.7 | 0.8 | 0.9 |
|---|---|---|---|---|---|---|---|---|---|---|
| HAN | reweight | 0.541 ± 0.011 | 0.551 ± 0.016 | 0.556 ± 0.016 | 0.560 ± 0.012 | 0.569 ± 0.010 | 0.567 ± 0.018 | 0.569 ± 0.014 | 0.574 ± 0.010 | **0.581 ± 0.017** |
| | embed_sm | 0.544 ± 0.010 | 0.543 ± 0.019 | 0.545 ± 0.012 | 0.549 ± 0.011 | 0.555 ± 0.014 | 0.555 ± 0.007 | 0.560 ± 0.017 | 0.558 ± 0.014 | 0.571 ± 0.012 |
| | em_smote | 0.547 ± 0.021 | 0.553 ± 0.015 | 0.556 ± 0.016 | 0.558 ± 0.012 | 0.568 ± 0.014 | 0.568 ± 0.012 | 0.572 ± 0.019 | 0.566 ± 0.018 | 0.568 ± 0.013 |
| | *HetGSM* | **0.553 ± 0.010** | **0.561 ± 0.017** | **0.562 ± 0.016** | **0.571 ± 0.021** | 0.570 ± 0.024 | **0.573 ± 0.015** | **0.575 ± 0.017** | **0.576 ± 0.019** | 0.579 ± 0.018 |
| MAGNN | reweight | 0.536 ± 0.013 | 0.544 ± 0.020 | 0.549 ± 0.016 | 0.540 ± 0.010 | 0.541 ± 0.021 | 0.550 ± 0.012 | 0.544 ± 0.008 | 0.542 ± 0.010 | 0.542 ± 0.013 |
| | embed_sm | 0.505 ± 0.002 | 0.510 ± 0.011 | 0.510 ± 0.010 | 0.514 ± 0.008 | 0.512 ± 0.007 | 0.518 ± 0.012 | 0.523 ± 0.007 | 0.535 ± 0.010 | 0.543 ± 0.008 |
| | em_smote | 0.525 ± 0.022 | 0.521 ± 0.006 | 0.517 ± 0.012 | 0.518 ± 0.015 | 0.523 ± 0.020 | 0.522 ± 0.015 | 0.520 ± 0.014 | 0.513 ± 0.010 | 0.523 ± 0.013 |
| | *HetGSM* | **0.539 ± 0.017** | **0.556 ± 0.015** | **0.556 ± 0.012** | **0.555 ± 0.015** | **0.553 ± 0.020** | **0.559 ± 0.014** | **0.559 ± 0.018** | **0.558 ± 0.015** | **0.553 ± 0.016** |
| HGNN | reweight | 0.532 ± 0.009 | 0.533 ± 0.022 | 0.533 ± 0.014 | 0.541 ± 0.016 | 0.546 ± 0.013 | 0.542 ± 0.019 | 0.541 ± 0.015 | 0.548 ± 0.024 | **0.549 ± 0.019** |
| | embed_sm | 0.525 ± 0.015 | 0.532 ± 0.019 | 0.533 ± 0.010 | 0.542 ± 0.021 | 0.547 ± 0.017 | 0.546 ± 0.028 | 0.546 ± 0.021 | 0.547 ± 0.017 | 0.546 ± 0.018 |
| | em_smote | 0.530 ± 0.010 | 0.531 ± 0.020 | 0.528 ± 0.016 | 0.542 ± 0.011 | **0.551 ± 0.016** | 0.547 ± 0.014 | 0.538 ± 0.018 | 0.544 ± 0.018 | 0.547 ± 0.023 |
| | *HetGSM* | **0.534 ± 0.014** | **0.532 ± 0.013** | **0.537 ± 0.020** | **0.543 ± 0.014** | 0.549 ± 0.023 | **0.548 ± 0.017** | **0.551 ± 0.023** | **0.551 ± 0.014** | 0.540 ± 0.028 |

**Table D.2.** AUC scores for HetGSMOTE (across all base models) under varying imbalance ratios on the IMDb dataset.

| # | Settings | 0.1 | 0.2 | 0.3 | 0.4 | 0.5 | 0.6 | 0.7 | 0.8 | 0.9 |
|---|---|---|---|---|---|---|---|---|---|---|
| HAN | reweight | 0.658 ± 0.023 | 0.679 ± 0.031 | 0.684 ± 0.023 | 0.688 ± 0.028 | 0.703 ± 0.027 | 0.706 ± 0.021 | 0.704 ± 0.029 | 0.708 ± 0.025 | 0.708 ± 0.021 |
| | embed_sm | 0.662 ± 0.034 | 0.675 ± 0.033 | 0.683 ± 0.028 | 0.688 ± 0.022 | 0.704 ± 0.026 | 0.699 ± 0.026 | 0.704 ± 0.024 | 0.706 ± 0.021 | 0.716 ± 0.015 |
| | em_smote | 0.664 ± 0.022 | 0.680 ± 0.025 | **0.697 ± 0.026** | 0.699 ± 0.016 | **0.705 ± 0.032** | 0.707 ± 0.018 | **0.709 ± 0.023** | 0.713 ± 0.025 | 0.713 ± 0.027 |
| | *HetGSM* | **0.667 ± 0.020** | **0.685 ± 0.024** | 0.692 ± 0.019 | **0.705 ± 0.022** | 0.702 ± 0.023 | **0.713 ± 0.016** | 0.705 ± 0.020 | **0.713 ± 0.023** | **0.717 ± 0.024** |
| MAGNN | reweight | 0.653 ± 0.025 | 0.659 ± 0.030 | 0.672 ± 0.018 | 0.671 ± 0.026 | 0.649 ± 0.017 | 0.640 ± 0.022 | 0.633 ± 0.018 | 0.629 ± 0.021 | 0.628 ± 0.019 |
| | embed_sm | 0.597 ± 0.013 | 0.600 ± 0.019 | 0.607 ± 0.020 | 0.608 ± 0.018 | 0.614 ± 0.021 | 0.608 ± 0.029 | 0.616 ± 0.014 | 0.657 ± 0.034 | 0.667 ± 0.011 |
| | em_smote | 0.638 ± 0.030 | 0.673 ± 0.022 | 0.666 ± 0.011 | 0.676 ± 0.025 | 0.670 ± 0.022 | 0.675 ± 0.023 | 0.682 ± 0.023 | 0.681 ± 0.028 | 0.679 ± 0.033 |
| | *HetGSM* | **0.655 ± 0.019** | **0.673 ± 0.022** | **0.676 ± 0.024** | **0.687 ± 0.020** | **0.686 ± 0.025** | **0.694 ± 0.020** | **0.694 ± 0.022** | **0.694 ± 0.018** | **0.688 ± 0.022** |
| HGNN | reweight | 0.647 ± 0.017 | 0.659 ± 0.035 | 0.670 ± 0.024 | 0.679 ± 0.021 | 0.684 ± 0.018 | 0.684 ± 0.021 | **0.695 ± 0.026** | 0.687 ± 0.021 | **0.696 ± 0.021** |
| | embed_sm | 0.645 ± 0.026 | 0.659 ± 0.026 | 0.669 ± 0.027 | 0.678 ± 0.020 | 0.687 ± 0.027 | 0.683 ± 0.030 | 0.688 ± 0.025 | 0.689 ± 0.018 | 0.688 ± 0.019 |
| | em_smote | 0.647 ± 0.027 | 0.662 ± 0.036 | 0.670 ± 0.029 | 0.681 ± 0.010 | 0.688 ± 0.023 | 0.691 ± 0.019 | 0.682 ± 0.024 | 0.684 ± 0.020 | 0.686 ± 0.027 |
| | *HetGSM* | **0.650 ± 0.025** | **0.661 ± 0.032** | **0.670 ± 0.021** | **0.681 ± 0.020** | **0.688 ± 0.022** | **0.693 ± 0.020** | 0.682 ± 0.022 | **0.693 ± 0.021** | 0.694 ± 0.021 |

complementing the accuracy-based comparison in Table 4. The trends remain fully aligned with those in the main text: HAN continues to perform best among the base models, and HetGSMOTE consistently surpasses all baselines across every architecture. These results confirm that the effectiveness of HetGSMOTE is preserved regardless of the evaluation metric or underlying model.

## D.2 Variation Across Datasets

Tables D.3 and D.4 extend the dataset-wise comparison from Table 5. As observed earlier with accuracy, heterogeneous datasets with uneven node-type distributions still show greater metric variability. Despite this, HetGSMOTE maintains its advantage over the most competitive baselines across all datasets.

The extended F1 and AUC results further reinforce two core findings: (1) HetGSMOTE performs competitively or better in the majority of cases, and (2) the performance gains are more pronounced under severe imbalance conditions.

## D.3 Ablation with Imbalance Ratio

Tables D.5 and D.6 provide the extended F1 and AUC results for the ablation study on imbalance ratios using the HAN base model, complementing the accuracy comparison in Table 6. As with the main findings, the full HetGSMOTE variant, incorporating both pre-training and synthetic soft edges, retains its superior performance across most imbalance levels and secures the top score in the majority of test settings.

**Table D.3.** F1 scores of HetGSMOTE using the HAN base model across multiple datasets under varying imbalance ratios.

| # | Settings | 0.1 | 0.2 | 0.3 | 0.4 | 0.5 | 0.6 | 0.7 | 0.8 | 0.9 |
|---|---|---|---|---|---|---|---|---|---|---|
| IMDb | reweight | 0.541 ± 0.011 | 0.551 ± 0.016 | 0.556 ± 0.016 | 0.560 ± 0.012 | 0.569 ± 0.010 | 0.567 ± 0.018 | 0.569 ± 0.014 | 0.574 ± 0.010 | **0.581 ± 0.017** |
| | embed_sm | 0.544 ± 0.010 | 0.543 ± 0.019 | 0.545 ± 0.012 | 0.549 ± 0.011 | 0.555 ± 0.014 | 0.555 ± 0.007 | 0.560 ± 0.017 | 0.558 ± 0.014 | 0.571 ± 0.012 |
| | em_smote | 0.547 ± 0.021 | 0.553 ± 0.015 | 0.556 ± 0.016 | 0.558 ± 0.012 | 0.568 ± 0.014 | 0.568 ± 0.012 | 0.572 ± 0.019 | 0.566 ± 0.018 | 0.568 ± 0.013 |
| | *HetGSM* | **0.553 ± 0.010** | **0.561 ± 0.017** | **0.562 ± 0.016** | **0.571 ± 0.021** | **0.570 ± 0.024** | **0.573 ± 0.015** | **0.575 ± 0.017** | **0.576 ± 0.019** | 0.579 ± 0.018 |
| AMINER | reweight | 0.955 ± 0.008 | 0.958 ± 0.011 | 0.958 ± 0.010 | 0.962 ± 0.008 | 0.961 ± 0.009 | 0.961 ± 0.010 | 0.963 ± 0.011 | 0.963 ± 0.012 | 0.965 ± 0.011 |
| | embed_sm | 0.955 ± 0.009 | 0.957 ± 0.010 | 0.962 ± 0.012 | 0.963 ± 0.010 | 0.961 ± 0.012 | **0.964 ± 0.013** | 0.960 ± 0.010 | **0.965 ± 0.009** | 0.965 ± 0.010 |
| | em_smote | 0.960 ± 0.011 | 0.962 ± 0.009 | 0.962 ± 0.012 | 0.962 ± 0.011 | 0.964 ± 0.009 | 0.963 ± 0.011 | 0.964 ± 0.011 | 0.965 ± 0.010 | 0.968 ± 0.011 |
| | *HetGSM* | **0.960 ± 0.010** | **0.965 ± 0.014** | **0.965 ± 0.009** | **0.963 ± 0.009** | **0.965 ± 0.010** | 0.963 ± 0.008 | **0.965 ± 0.012** | 0.963 ± 0.009 | **0.968 ± 0.009** |
| PubMed | reweight | 0.167 ± 0.019 | 0.168 ± 0.019 | 0.153 ± 0.013 | **0.154 ± 0.015** | 0.154 ± 0.017 | 0.158 ± 0.017 | **0.164 ± 0.016** | 0.156 ± 0.016 | 0.153 ± 0.009 |
| | embed_sm | 0.149 ± 0.015 | 0.149 ± 0.015 | 0.164 ± 0.011 | 0.151 ± 0.012 | **0.158 ± 0.006** | 0.153 ± 0.012 | 0.154 ± 0.015 | 0.153 ± 0.016 | **0.159 ± 0.019** |
| | em_smote | 0.154 ± 0.021 | 0.154 ± 0.021 | 0.157 ± 0.016 | 0.153 ± 0.013 | 0.154 ± 0.013 | **0.160 ± 0.012** | 0.156 ± 0.017 | 0.146 ± 0.015 | 0.154 ± 0.018 |
| | *HetGSM* | **0.171 ± 0.017** | **0.171 ± 0.017** | **0.169 ± 0.016** | 0.144 ± 0.017 | 0.158 ± 0.020 | 0.146 ± 0.012 | 0.158 ± 0.017 | **0.156 ± 0.017** | 0.147 ± 0.010 |
| DBLP | reweight | 0.717 ± 0.036 | 0.771 ± 0.034 | 0.787 ± 0.035 | 0.796 ± 0.047 | **0.804 ± 0.032** | **0.808 ± 0.019** | 0.795 ± 0.030 | 0.807 ± 0.041 | 0.799 ± 0.034 |
| | embed_sm | 0.703 ± 0.022 | 0.734 ± 0.032 | 0.750 ± 0.022 | 0.765 ± 0.031 | 0.777 ± 0.140 | 0.785 ± 0.016 | 0.778 ± 0.025 | 0.783 ± 0.020 | 0.803 ± 0.029 |
| | em_smote | 0.722 ± 0.042 | 0.757 ± 0.034 | 0.771 ± 0.022 | 0.782 ± 0.036 | 0.790 ± 0.024 | 0.796 ± 0.020 | 0.772 ± 0.036 | 0.789 ± 0.028 | 0.781 ± 0.024 |
| | *HetGSM* | **0.732 ± 0.028** | **0.774 ± 0.041** | **0.788 ± 0.029** | **0.796 ± 0.032** | 0.800 ± 0.032 | 0.804 ± 0.027 | **0.808 ± 0.035** | **0.807 ± 0.031** | **0.815 ± 0.026** |

**Table D.4.** AUC scores of HetGSMOTE using the HAN base model across multiple datasets under varying imbalance ratios.

| # | Settings | 0.1 | 0.2 | 0.3 | 0.4 | 0.5 | 0.6 | 0.7 | 0.8 | 0.9 |
|---|---|---|---|---|---|---|---|---|---|---|
| IMDb | reweight | 0.658 ± 0.023 | 0.679 ± 0.031 | 0.684 ± 0.023 | 0.688 ± 0.028 | 0.703 ± 0.027 | 0.706 ± 0.021 | 0.704 ± 0.029 | 0.708 ± 0.025 | 0.708 ± 0.021 |
| | embed_sm | 0.662 ± 0.034 | 0.675 ± 0.033 | 0.683 ± 0.028 | 0.688 ± 0.022 | 0.704 ± 0.026 | 0.699 ± 0.026 | 0.704 ± 0.024 | 0.706 ± 0.021 | 0.716 ± 0.015 |
| | em_smote | 0.664 ± 0.022 | 0.680 ± 0.025 | **0.697 ± 0.026** | 0.699 ± 0.016 | **0.705 ± 0.032** | 0.707 ± 0.018 | **0.709 ± 0.023** | 0.713 ± 0.025 | 0.713 ± 0.027 |
| | *HetGSM* | **0.667 ± 0.020** | **0.685 ± 0.024** | 0.692 ± 0.019 | **0.705 ± 0.022** | 0.702 ± 0.023 | **0.713 ± 0.016** | 0.705 ± 0.020 | **0.713 ± 0.023** | **0.717 ± 0.024** |
| AMINER | reweight | 0.994 ± 0.003 | 0.996 ± 0.002 | 0.996 ± 0.002 | 0.997 ± 0.002 | 0.997 ± 0.001 | 0.997 ± 0.002 | 0.996 ± 0.003 | 0.997 ± 0.002 | 0.998 ± 0.001 |
| | embed_sm | 0.994 ± 0.003 | 0.996 ± 0.002 | 0.996 ± 0.002 | 0.997 ± 0.001 | 0.997 ± 0.002 | 0.997 ± 0.001 | 0.997 ± 0.001 | 0.998 ± 0.001 | 0.997 ± 0.001 |
| | em_smote | 0.995 ± 0.002 | 0.996 ± 0.003 | 0.997 ± 0.002 | 0.997 ± 0.002 | **0.998 ± 0.001** | **0.998 ± 0.001** | 0.997 ± 0.002 | **0.998 ± 0.001** | 0.997 ± 0.001 |
| | *HetGSM* | **0.996 ± 0.002** | **0.997 ± 0.002** | **0.998 ± 0.001** | **0.997 ± 0.001** | 0.997 ± 0.001 | 0.997 ± 0.002 | **0.997 ± 0.001** | 0.997 ± 0.002 | **0.998 ± 0.001** |
| PubMed | reweight | 0.536 ± 0.021 | 0.536 ± 0.021 | 0.514 ± 0.020 | 0.513 ± 0.016 | 0.505 ± 0.023 | **0.521 ± 0.023** | 0.521 ± 0.037 | **0.529 ± 0.020** | 0.519 ± 0.025 |
| | embed_sm | 0.515 ± 0.026 | 0.516 ± 0.026 | 0.523 ± 0.032 | 0.516 ± 0.025 | 0.499 ± 0.020 | 0.513 ± 0.027 | 0.521 ± 0.023 | 0.517 ± 0.017 | 0.525 ± 0.020 |
| | em_smote | 0.532 ± 0.032 | 0.533 ± 0.032 | 0.513 ± 0.021 | **0.517 ± 0.021** | **0.529 ± 0.020** | 0.511 ± 0.020 | 0.509 ± 0.021 | 0.516 ± 0.025 | 0.520 ± 0.011 |
| | *HetGSM* | **0.538 ± 0.014** | **0.539 ± 0.015** | **0.526 ± 0.027** | 0.513 ± 0.023 | 0.506 ± 0.029 | 0.508 ± 0.017 | **0.517 ± 0.036** | 0.506 ± 0.025 | **0.530 ± 0.021** |
| DBLP | reweight | 0.891 ± 0.024 | 0.917 ± 0.017 | 0.926 ± 0.023 | 0.940 ± 0.022 | **0.946 ± 0.014** | 0.950 ± 0.008 | **0.952 ± 0.008** | 0.950 ± 0.007 | 0.943 ± 0.015 |
| | embed_sm | 0.880 ± 0.017 | 0.902 ± 0.019 | 0.916 ± 0.012 | 0.924 ± 0.016 | 0.932 ± 0.009 | 0.938 ± 0.007 | 0.931 ± 0.013 | 0.939 ± 0.010 | 0.945 ± 0.012 |
| | em_smote | 0.892 ± 0.018 | 0.916 ± 0.019 | 0.928 ± 0.013 | 0.935 ± 0.016 | 0.943 ± 0.013 | 0.942 ± 0.010 | 0.930 ± 0.019 | 0.941 ± 0.014 | 0.937 ± 0.011 |
| | *HetGSM* | **0.894 ± 0.029** | **0.929 ± 0.019** | **0.936 ± 0.018** | **0.942 ± 0.013** | 0.942 ± 0.014 | **0.952 ± 0.015** | 0.942 ± 0.014 | 0.945 ± 0.020 | **0.952 ± 0.014** |

**Table D.5.** F1 scores of HetGSMOTE variants (HAN base model) under different imbalance ratios on the IMDb dataset.

| Settings | 0.1 | 0.2 | 0.3 | 0.4 | 0.5 |
|---|---|---|---|---|---|
| *GSM_base* | 0.545 ± 0.014 | 0.552 ± 0.017 | 0.562 ± 0.011 | 0.556 ± 0.022 | 0.569 ± 0.021 |
| *GSM_S* | 0.547 ± 0.021 | 0.553 ± 0.016 | 0.557 ± 0.017 | 0.557 ± 0.018 | 0.567 ± 0.018 |
| *GSM_PT* | 0.551 ± 0.009 | 0.559 ± 0.020 | **0.565 ± 0.020** | 0.565 ± 0.012 | **0.570 ± 0.017** |
| *HetGSM* | **0.553 ± 0.010** | **0.561 ± 0.017** | 0.562 ± 0.016 | **0.571 ± 0.021** | **0.570 ± 0.024** |

**Table D.6.** AUC scores of HetGSMOTE variants (HAN base model) under different imbalance ratios on the IMDb dataset.

| Settings | 0.1 | 0.2 | 0.3 | 0.4 | 0.5 |
|---|---|---|---|---|---|
| *GSM_base* | 0.658 ± 0.023 | 0.679 ± 0.031 | 0.684 ± 0.023 | 0.688 ± 0.028 | 0.703 ± 0.027 |
| *GSM_S* | 0.662 ± 0.034 | 0.675 ± 0.033 | 0.683 ± 0.028 | 0.688 ± 0.022 | 0.704 ± 0.026 |
| *GSM_PT* | 0.664 ± 0.022 | 0.680 ± 0.025 | **0.697 ± 0.026** | 0.699 ± 0.016 | **0.705 ± 0.032** |
| *HetGSM* | **0.667 ± 0.020** | **0.685 ± 0.024** | 0.692 ± 0.019 | **0.705 ± 0.022** | 0.702 ± 0.023 |

