# OpenReview forum: "HetGSMOTE: Oversampling for Heterogeneous Graphs"
_NLDL.org/2026/Conference — NLDL 2026 Spotlight_

### Official Review · Reviewer_Bmww · 2025-10-04
**Technically Sound and Well-Evaluated Framework for Addressing Class Imbalance in Heterogeneous Graphs**

**Rating:** 4
**Confidence:** 3
**Final Rating:** 4
**Final Confidence:** 3

**Summary:**

The paper introduces HetGSMOTE, a novel oversampling framework designed to handle class imbalance in heterogeneous graph neural networks (HetGs). Unlike prior SMOTE-based methods (GraphSMOTE) that target homogeneous graphs, HetGSMOTE integrates node-type, edge-type, and metapath information to generate synthetic minority samples while preserving structural and semantic relationships. The framework consists of four components, including heterogeneous graph encoding, SMOTE-based oversampling in the learned embedding space, neural edge generation, and final node classification. Comprehensive experiments on multiple benchmark datasets (AMiner, DBLP, IMDb, PubMed) and across various base HGNNs (HAN, MAGNN, HetGNN) demonstrate that HetGSMOTE consistently outperforms traditional oversampling and reweighting baselines, particularly under severe imbalance conditions. The approach shows robustness and flexibility across graph types and imbalance ratios.

**Strengths:**

1. The paper addresses a critical challenge of class imbalance in heterogeneous graphs and presents a concrete, technically sound solution.
2. The design is logical and modular. The separation of encoding, oversampling, and edge generation is well-motivated.
3. The experiments are thorough. The inclusion of ablation studies and parameter sensitivity analyses adds credibility to the results.

**Weaknesses:**

1. In the related work section, the authors mention recent approaches such as FincGAN and BARE, which specifically address class imbalance in heterogeneous graphs. However, the experimental evaluation does not include comparisons with these methods, instead restricting baselines to traditional and SMOTE-based techniques.
2. The paper is primarily empirical and would benefit from a more detailed theoretical or analytical discussion explaining why SMOTE-based oversampling effectively improves minority-class representation, particularly in heterogeneous graph settings where node and edge types introduce complex structural dependencies.

**Final Justification:**

The paper is well-written, technically sound, and supported by a solid experimental evaluation.

**Justification:**

Overall, I lean toward accepting this paper. It addresses an important problem of class imbalance in heterogeneous graphs, and provides a technically sound and well-structured solution through the proposed HetGSMOTE framework. The design is logical and modular, integrating heterogeneous graph encoding, embedding-space oversampling, and neural edge generation in a cohesive manner. The experimental evaluation is comprehensive, covering multiple datasets, imbalance ratios, and backbone architectures, with ablation and sensitivity analyses that convincingly support the method’s effectiveness.

While the work is largely empirical, it is executed carefully and produces consistent improvements across benchmarks. The main limitations are the lack of direct comparison with recently proposed heterogeneous graph oversampling methods such as FincGAN and BARE, and the absence of theoretical discussion explaining why SMOTE-based oversampling remains effective in heterogeneous settings. Nonetheless, these weaknesses do not substantially diminish the contribution.

---

> ### Author Rebuttal · Authors · 2025-10-18
>
> We sincerely thank the reviewer for the detailed and positive assessment of our work. We are glad that the reviewer finds HetGSMOTE to be technically sound, logically designed, and empirically effective across multiple datasets and HGNN backbones. We also appreciate the recognition of our modular framework, ablation studies, and parameter sensitivity analyses.  We will address the concerns raised by the reviewer, following the same order and numbering as in the review.
>
> 1. We appreciate the reviewer's observation. In this study, our primary focus is on extending SMOTE-based oversampling to heterogeneous graphs due to its simplicity, interpretability, and relatively low computational cost. Methods like FinC-GAN and BARE rely on different generative mechanisms and, therefore, are not directly comparable within the SMOTE framework. However, these techniques can potentially be combined with HetGSMOTE to further enhance minority representation, which we plan to explore as a future research direction.
>
> 2. We thank the reviewer for this suggestion. Our work focuses on the empirical evaluation of HetGSMOTE, demonstrating its effectiveness across multiple datasets and backbones. The theoretical rationale for SMOTE-based oversampling is already well-established in prior literature for homogeneous and low-dimensional feature spaces, providing a solid foundation for its use in heterogeneous graphs. Extending these analytical results to fully characterize the interplay of node- and edge-type dependencies is an interesting direction and will be considered in future work.

---

### Official Review · Reviewer_wvP5 · 2025-10-07
**Weaknesses outweigh strengths**

**Rating:** 2
**Confidence:** 4
**Final Rating:** 4
**Final Confidence:** 3

**Summary:**

The paper addresses an important problem with relevance to real-world applications and presents a natural extension to existing work in attempt to fulfill a more difficult research gap.

**Strengths:**

1. The paper addresses an important problem with relevance to real-world applications and presents a natural extension to existing work in attempt to fulfill a more difficult research gap.

2. A variety of base HGNNs, to which HetGSMOTE is applied, is selection including simple, attention-based and meta-path based.

3. The illustrations convey the process outline clearly.

**Weaknesses:**

1. As mentioned on line 428, AUC is a better evaluation metric than accuracy in cases of class imbalance. Yet, only one set of results report AUC in Table 3 for one dataset. All other results for other datasets in Tables 2, 4 and 5 report accuracy.

2. Other techniques specialized for class imbalance in HetGs, as mentioned on lines 148-150, are not evaluated against HetGSMOTE, or are they complementary to HetGSMOTE and can be used in combination?

3. On line 176, it is mentioned that $F^t[v_j,:]$ denotes the feature vector for node $v_j$ of type $t$, and on line 272 it says that $G^t[v_j,:]$ denotes the aggregated embedding of type $t$ neighbors for node $v_j$. This implies that for all types of $t' \in \tau$ neighbors of $v_j$ there exists an $G^{t'}[v_j,:]$ as used in Eq. 2. Then, what does $F^{t'}[v_j,:]$ denote in Eq. 2? The same question arises for the generalized forumation on line 278.

4. Notation is used (section 3) before definition (section 4 and appendix) and is overlapping and unclear (for e.g. Y is used both for  labels on line 182 and for a feature vector in definition of SMOTE after line 157). Again, on lines 377 and 388, $\beta$ and $\lambda$ serve the same purpose but different notation is used in both sections.

5. On line 248, do 'attribute' and 'feature' refer to the same thing? If so , then on line 253, what is the difference between $d$ and $d_i$, as $d$ is defined to be the dimension of the feature vector for all note types in section 4? Due to this, it is unclear how the content aggregation step really works.

6. Metric used in Table 6 and Table 7 not mentioned.

7. missing citations on line 118.

8. On line 146, the authors mention that GSMOTE has found limited applications in graph-based learning(assuming even for homogenous graphs?). What are the limitations of GSMOTE that restrict its applicability and does HetGSMOTE overcome them for heterogenous graphs? It would make more sense that HetGSMOTE is developed to cater to heterogenous graphs, given the applicability GSMOTE has found in homogenous graphs already. This raises doubts about the motivation for (and applicability of) HetGSMOTE.

**Final Justification:**

The authors ave addressed my questions and concerns from the initial review during the rebuttal satisfactorily. Incorporating the  provided clarifications in the paper will greatly improve its presentation, readability, and clarity of thought. Therefore, I now lean towards acceptance.

**Justification:**

While the technical idea to solve an important problem may be interesting, I believe its execution in this paper leaves more to be desired and should be revisited in terms of quality of presentation and evaluation, clarity of thought, and significance of the work as discussed above in the weaknesses section and raises several confusing questions for the reader.

---

> ### Author Rebuttal · Authors · 2025-10-18
>
> We thank the reviewer for their careful reading and constructive feedback. We appreciate the recognition of HetGSMOTE's relevance to real-world heterogeneous graph applications, the diversity of HGNN backbones evaluated, and the clarity of illustrations. Below, we address concerns raised in each point by the reviewer, following the same order and numbering as in the review.
>
> 1. We agree that AUC provides a more nuanced evaluation under class imbalance. However, the reported accuracies in Tables 2-5 are macro-averaged, i.e., computed by averaging per-class accuracies, which mitigates imbalance bias. Our primary goal is to ensure consistent comparison between baseline models and HetGSMOTE, rather than emphasizing absolute performance on a single metric. Due to space constraints, we included detailed AUC and macro-F1 results for only one representative dataset in the main paper; for completeness, we plan to include full AUC and macro-F1 results for all datasets in the revised Appendix. Preliminary results across all metrics indicate similar improvements, reinforcing the claims made in the main paper.
>
> 2. We acknowledge the reviewer's suggestion to include heterogeneity-specific imbalance techniques such as FinC-GAN and BARE. Our focus in this work is on SMOTE-based oversampling methods, chosen for their simplicity, interpretability, and lower computational overhead compared to adversarial or autoencoder-based approaches. FinC-GAN and BARE employ substantially different generative paradigms and are not directly comparable to SMOTE-based techniques. We aimed to systematically extend the SMOTE family to heterogeneous graphs, ensuring fair comparisons among methods with similar assumptions and complexity. We note that techniques like BARE and FinC-GAN, being non-SMOTE-based, could be used in combination with HetGSMOTE if desired. We shall add this as a potential future direction in Section 8 (lines 570-573).
>
> 3. We thank the reviewer for pointing this out. The attention weights in Eq. (2) will be updated as follows:
>
>     $$
>     \alpha_{v_j}^{t'} = \frac{\exp\Big(\sigma\big(W_a \cdot (G^{t'}[v_j,:] \oplus F^t[v_j,:])\big)\Big)}{\sum_{t' \in \mathcal{T}} \exp\Big(\sigma\big(W_a \cdot (G^{t'}[v_j,:] \oplus F^t[v_j,:])\big)\Big)}
>     $$
>
>     Here, $F^t[v_j,:]$ is the content-aggregated feature matrix for node $v_j$ of its own type $t$ (from Eq. (1)), and $G^{t'}[v_j,:]$ is the neighbor-aggregated embedding for neighbors of type $t'$. The numerator uses the neighbor type $t'$ corresponding to that attention weight, and the denominator sums over all neighbor types $t' \in \mathcal{T}$ to ensure proper normalization. We acknowledge that the previous notation was ambiguous and will revise the manuscript to clearly distinguish $F^t$ (content features) from $G^{t'}$ (neighbor embeddings), and to clarify that attention weights are computed after obtaining both matrices.
>
>
>
> 4. We will move the equations to follow their respective definitions, leaving only the general equation in the background and referencing it there. All instances of ambiguous symbols will be corrected: $Y$ will consistently denote labels, and the SMOTE equation will be updated accordingly. We will retain $\lambda$ as the standard notation for the hyperparameter in the loss function to ensure consistency throughout the manuscript.
>
> 5. Here, “attribute” and “feature” are used interchangeably. The term $d_i$ denotes the dimensionality of the $i$-th attribute matrix for nodes of type $t$, while $d$ represents the unified embedding dimension after projection. As shown in Eq.~(1), all attribute matrices $F_i^t \in \mathbb{R}^{n_t \times d_i}$ are concatenated and then passed through a linear layer $W_c \in \mathbb{R}^{d \times \sum_i d_i}$ followed by a non-linear activation to obtain $F^t[v_j, :] \in \mathbb{R}^d$. This ensures that heterogeneous attributes are projected into a common feature space before aggregation.
>
> 6. We thank the reviewer for pointing this out. The manuscript will be updated to explicitly mention the evaluation metric(s) used in Tables 6 and 7.
>
> 7. We thank the reviewer for pointing this out. The missing citations will be added in the revised manuscript to properly acknowledge prior work.
>
> 8. We note that the line refers to SMOTE-based methods in general, and GSMOTE is one such method designed only for homogeneous graphs, where all nodes and edges share the same type. These methods do not account for type-specific semantics, edge heterogeneity, or non-shared feature spaces. HetGSMOTE overcomes these limitations by performing type-aware embedding interpolation with separate latent subspaces per node type and edge-type-conditioned generation, ensuring semantic consistency in heterogeneous graphs.

---

### Official Review · Reviewer_UhiY · 2025-10-09

**Rating:** 5
**Confidence:** 4

**Summary:**

This paper considers the problem of node classifications for imbalanced datasets when using heterogeneous. This is an important problem as many times nodes labels can be severely imbalanced. The paper presents a new HetGSMOTE oversampling technique, that samples synthetic data points for the underrepresented classes. The paper compares their method against existing techniques of four datasets and finds that their method consistently has the best performance.

Finally, they do ablation studies that significantly help understand the technique.

**Strengths:**

The paper has many strengths.

1. To begin it is well written and easy to follow.

2. The paper presents the new methods clearly with lots of details. It clearly distinguishes it from prior work and states what is novel.

3. The paper conducts extensive experiments. The experiments support the papers argument that the technique works in improving performance.

4. The paper also presents ablation studies which provide insights into how to tune hyper parameters of their technique.

**Weaknesses:**

No significant weakness that would prevent publishing

**Justification:**

The paper is well written, on an important topic and provides clear advancement.

---

> ### Author Rebuttal · Authors · 2025-10-18
>
> We sincerely thank the reviewer for the positive evaluation and thoughtful comments. We appreciate the recognition of HetGSMOTE's clarity, novelty, and extensive experimental validation. We are glad that the reviewer found our ablation studies informative for understanding hyperparameter choices and model behavior. The feedback encourages us to further explore heterogeneous graph oversampling strategies, and we will continue refining HetGSMOTE in future work.

---

### Official Review · Reviewer_7G29 · 2025-10-10

**Rating:** 2
**Confidence:** 3

**Summary:**

The paper introduces HetGSMOTE, an oversampling method for heterogeneous graphs. It builds a type-aware embedding space with a base HGNN, creates synthetic minority nodes via SMOTE-style interpolation, and learns per edge type generators to attach synthetic nodes to the graph. The classifier is trained on the augmented graph. Experiments across several datasets and backbones show improvements in many settings, especially under heavy imbalance, with ablations on oversampling ratios and components. The idea is practical and easy to plug into standard pipelines, but several modeling and evaluation choices limit confidence in the strength and generality of the gains.

**Strengths:**

- Addresses a relevant and underexplored problem for heterogeneous graphs.
- Clear, modular pipeline that practitioners can reproduce and integrate into existing HGNN backbones.
- Joint training of the classifier with an edge generator is a sensible way to couple labels and structure.
- Sensitivity studies and component ablations provide useful guidance on oversampling ratios and design choices.
 - Broad compatibility with common hetero backbones and multiple datasets increases practical impact.

**Weaknesses:**

- Edge generator training appears to under-penalize false positives, risking overly dense or noisy synthetic connectivity.
- Potential label leakage in SMOTE neighbor selection if the pool is not strictly limited to training-labeled nodes of the minority class.
- Train test mismatch if synthetic nodes and edges are present during training but removed at evaluation.
- Improvements are modest or within variance in several settings, with mixed results on some datasets.
- Missing comparisons to hetero-specific imbalance methods and limited analysis of structural fidelity around synthetic nodes.

**Justification:**

HetGSMOTE is well motivated and the design is clean and practical. The empirical gains are promising in harder imbalance regimes, and the method integrates smoothly with standard backbones. However, the current edge modeling raises concerns about spurious connectivity, the evaluation does not clearly rule out leakage or train test mismatch effects, and several improvements are small relative to variance. Stronger negatives-aware edge training, explicit guarantees on label scope, comparisons to hetero-specific baselines, and structural diagnostics around synthetic nodes would make the case more convincing.

---

> ### Author Rebuttal · Authors · 2025-10-18
>
> We sincerely thank the reviewer for the constructive and detailed feedback. We appreciate the recognition of HetGSMOTE's motivation, modular design, and empirical robustness across multiple heterogeneous graph benchmarks. Below, we address the reviewer's main concerns and clarify several methodological and evaluation aspects. We have provided our rebuttal corresponding to each point raised by the reviewer, following the same order and numbering as in the review.
>
> 1. We agree that dense or noisy connectivity is a potential concern. In our setup, false positives are not explicitly penalized but are implicitly controlled through the edge loss and a fixed threshold of 0.5 applied to predicted edge probabilities. This follows the design choice in GraphSMOTE, where such a thresholding mechanism effectively mitigates over-connectivity while keeping the generator lightweight and stable.
>
> 2. We confirm that testing labels are strictly hidden until evaluation. The SMOTE neighborhood selection is performed exclusively within the training set and restricted to minority class nodes. This ensures that no label information from validation or test splits leaks into the synthetic node generation process.
>
> 3. The motivation for oversampling in class-imbalance problems is to provide sufficient minority-class representation for effective learning during training. As stated in lines 182-186, synthetic nodes are introduced only at the training stage to address data scarcity. During evaluation, the model is tested solely on the original test set without synthetic nodes or edges, which is standard practice in oversampling-based methods. This separation ensures that the reported performance reflects the model's generalization to real data.
>
> 4. We acknowledge that some results show modest improvements. However, HetGSMOTE consistently achieves comparatively strong performance under extreme imbalance ratios ($<0.5$), where the method is most relevant. We intentionally report extended results to present a complete and transparent evaluation, including limitations. Notably, prior works such as GraphSMOTE also report similar levels of variance and incremental improvements for analogous oversampling setups. The IMDb dataset, used for the main experiments (lines 480-485), is relatively small and exhibits a balanced distribution of node types. In contrast, other heterogeneous graph datasets have imbalanced distributions across different node types, leading to greater fluctuations in performance. Such variability is expected in heterogeneous settings, where class imbalance interacts with structural imbalance across node categories. We will clarify this dataset-specific imbalance distinction explicitly in the revised manuscript for completeness.
>
> 5. For the scope of this study, we focus on SMOTE-based oversampling strategies and their extension to heterogeneous graphs, aiming for a direct conceptual comparison to methods like GraphSMOTE rather than hetero-specific techniques that rely on substantially different architectures like GANs. Despite not explicitly analyzing structural fidelity, the improved performance across heterogeneous datasets indicates that the generated synthetic nodes preserve meaningful structural consistency. We will explore a detailed structural fidelity and theoretical analysis in future work to further support the observed empirical trends.

---

### Meta-Review · Area_Chair_DFWy · 2025-10-30

**Recommendation:** Accept (Poster)
**Confidence:** 4

**Metareview:**

This paper introduces HetGSMOTE, an oversampling method for imbalanced classes in heterogenous graphs. The method creates synthetic samples for the underrepresented classes by applying SMOTE oversampling in the learned latent space, taking into account node types.

The reviewers agree on the importance of the problem studied. The approach chosen by the authors extends existing methods to be suitable for the heterogenous graph setting in a natural way. The experimental part is solid, and the results show an improvement, especially in the case of a strong imbalance.

One of the reviewers was concerned with the risk of under-penalizing the false positives. This risk is, according to the authors, mitigated by the edge loss combined with a thresholding. This mitigation is inspired from GraphSMOTE.

Reviewers pointed a number of things to be improved in the paper:
- since the main goal of the method is to deal with class imbalance, the AUC metric should be privileged in the results
- lack of direct comparison with other non-SMOTE based methods.
- some notation issues
- study is mainly empirical and is lacking deeper theoretical analysis

Reviewers agree on the interest as well as the novelty of the method presented and the positive performance improvements it brings. Despite its mainly empirical nature, the method proposed in the paper is a natural extension of existing methods. The authors have clarified the weak points during the rebuttal, I therefore propose to accept the paper.

---

### Decision · Program_Chairs · 2025-11-05

**Decision:**

Accept (Spotlight)

**Comment:**

We recommend an oral and a poster presentation given the AC and reviewers recommendations.

A spotlight presentation refers to a poster selected for an oral highlight but not designated as a full oral presentation per the AC’s recommendation.